# Surface Motility Regulation of *Sinorhizobium fredii* HH103 by Plant Flavonoids and the NodD1, TtsI, NolR, and MucR1 Symbiotic Bacterial Regulators

**DOI:** 10.3390/ijms23147698

**Published:** 2022-07-12

**Authors:** Cynthia Alías-Villegas, Francisco Fuentes-Romero, Virginia Cuéllar, Pilar Navarro-Gómez, María J. Soto, José-María Vinardell, Sebastián Acosta-Jurado

**Affiliations:** 1Centro Andaluz de Biología del Desarrollo, CSIC/Junta de Andalucía, Departamento de Biología Molecular e Ingeniería Bioquímica, Universidad Pablo de Olavide, 41013 Seville, Spain; calivil@upo.es; 2Facultad de Biología, Departamento de Microbiología, Universidad de Sevilla, 41012 Sevilla, Spain; ffuentesr@us.es (F.F.-R.); pnavarro2@us.es (P.N.-G.); 3Estación Experimental del Zaidín, CSIC, Departamento de Biotecnología y Protección Ambiental, 18008 Granada, Spain; virginia.cuellar@eez.csic.es (V.C.); mariajose.soto@eez.csic.es (M.J.S.)

**Keywords:** *Sinorhizobium fredii*, surface motility, swarming, swimming, NodD, TtsI, NolR, slavonoids, T3SS, flagella

## Abstract

Bacteria can spread on surfaces to colonize new environments and access more resources. Rhizobia, a group of α- and β-*Proteobacteria*, establish nitrogen-fixing symbioses with legumes that rely on a complex signal interchange between the partners. Flavonoids exuded by plant roots and the bacterial transcriptional activator NodD control the transcription of different rhizobial genes (the so-called *nod* regulon) and, together with additional bacterial regulatory proteins (such as TtsI, MucR or NolR), influence the production of different rhizobial molecular signals. In *Sinorhizobium fredii* HH103, flavonoids and NodD have a negative effect on exopolysaccharide production and biofilm production. Since biofilm formation and motility are often inversely regulated, we have analysed whether flavonoids may influence the translocation of *S. fredii* HH103 on surfaces. We show that the presence of *nod* gene-inducing flavonoids does not affect swimming but promotes a mode of surface translocation, which involves both flagella-dependent and -independent mechanisms. This surface motility is regulated in a flavonoid-NodD1-TtsI-dependent manner, relies on the assembly of the symbiotic type 3 secretion system (T3SS), and involves the participation of additional modulators of the *nod* regulon (NolR and MucR1). To our knowledge, this is the first evidence indicating the participation of T3SS in surface motility in a plant-interacting bacterium. Interestingly, flavonoids acting as *nod*-gene inducers also participate in the inverse regulation of surface motility and biofilm formation, which could contribute to a more efficient plant colonisation.

## 1. Introduction

Rhizobia are soil bacteria able to establish a mutualistic relationship with plants belonging to the *Fabaceae* family. In this interaction, bacteria infect legume roots and colonise new organs, called nodules, formed by the plant. Inside nodules, rhizobia invade plant cells and differentiate into bacteroids able to fix atmospheric N_2_ into ammonia that is provided to the plant. In return, the plant supplies to the bacteria a carbon and energy source as well as an appropriate environment for nitrogen fixation [1,2].

The rhizobia–legume symbiotic interaction requires a coordinated, continuous and complex interchange of molecular signals between compatible rhizobia and legumes [3,4,5,6]. The nodulation process is initiated by the root exudation of a plethora of molecules which includes phenolic compounds called flavonoids [7]. The bacterial transcriptional regulator NodD interacts with appropriate flavonoids and, through its binding to conserved promoter sequences named *nod* boxes (NB) [8,9], activates the transcription of genes responsible for the production and secretion of bacterial molecular signals called Nod factors (NF, or lipochitooligosaccharides, LCOs). These NF are *N*-Acetyl-glucosamine oligosaccharides harbouring different molecular decorations and, when appropriate, are recognised by LysM receptors located on the root hair cells, triggering root infection and the formation of nodule primordia in the root cortex. In most legumes, bacteria reach the symbiotic cortical cells of nodules through tubular structures, called infection threads, that initiate in root hairs and connect the epidermal and the cortical layers of the root [3,10,11], although there are also intercellular infection strategies, such as crack entry [12].

Besides NF, other bacterial molecular signals may be important for an efficient symbiosis [4,13]. In some rhizobia, such as the broad host range *Sinorhizobium fredii* HH103, NodD also induces the expression and assembly of a type 3 secretion system (T3SS) through the activation of the expression of the *ttsI* transcriptional activator. The T3SS delivers effector proteins into host plant cells and modifies plant defence responses [5,14,15]. Rhizobial surface polysaccharides (RSP) can also act as signal molecules during the symbiotic interaction and/or as a protection against plant defences. Diverse RSP have been described in rhizobia, such as exopolysaccharides (EPS), the K-antigen capsular polysaccharide (KPS), lipopolysaccharides (LPS) and cyclic glucans (CGs) [4,13]. In *S. fredii* HH103, in addition to NodD1 (the master positive regulator of *nod* genes) and TtsI, other rhizobial transcriptional regulators, such as NolR (a global repressor of the *nod* regulon), MucR (another global regulator involved in multiple processes in both saprophytic and symbiotic stages), SyrM and NodD2 (both acting as activators and repressors of different genes of the *nod* regulon) play important roles in the fine tuning of the production of bacterial molecular signals involved in symbiosis [16,17,18,19,20,21].

In nature, microorganisms usually live within biofilms, complex microbial communities assembled in an extracellular matrix (containing bacterial EPS, as a key component) that adhere to a surface [22]. In their transition from soil saprophytes to legume endosymbionts, rhizobia attach and form biofilms on roots and root hairs prior to infecting legume roots [2]. Thus, the production of EPS is an important trait for colonisation of legume root surfaces by rhizobia. In fact, flavonoids exuded by legume roots have a positive effect on *R. leguminosarum* bv. *trifolii* EPS production [23]. However, in *S. fredii* HH103 *nod* gene-inducing flavonoids repress EPS production and biofilm formation in a NodD1- and SyrM-dependent manner [21,24], whereas NolR stimulates both processes [16].

Biofilm formation and surface motility are opposite behaviours exhibited by bacteria living on surfaces [22,25]. Rhizobial motility is not essential for nodulation and nitrogen fixation but can influence nodulation efficiency and competitiveness by approaching bacteria to potential infection sites and/or facilitating their progression inside infection threads [26,27,28]. Several studies have shown that rhizobial mutants affected in flagella synthesis exhibit partial defects in symbiotic performance such as reduced ability for root colonisation [29], delayed nodulation and/or reduced competitiveness for nodulating host legumes [25,30,31,32,33]. These results suggest that rhizobial motility may be an important trait for host root colonisation and infection. In fact, since rhizobia lack flagella inside infection threads, Fournier et al. [26] proposed that rhizobial penetration through these tubular structures is the result of the combination of surface translocation (flagella-independent) and cell division. On the other hand, in some studies genes related to flagellar biosynthesis have been shown to be strongly repressed inside nodules [34,35,36], in agreement with the non-motile state of bacteroids (the rhizobial differentiate forms present inside nodule cells and able to fix nitrogen). However, a recent work has shown that most flagellar assembly and basal body apparatus proteins are important in *R. leguminosarum* bacteroids and nodule bacteria in symbiosis with *Pisum sativum* [27].

Rhizobia can move using different mechanisms. The most studied is swimming, a flagella-driven motility that allows the movement of individual bacteria in liquid media. Nevertheless, like many other bacteria, rhizobia can also move on surfaces using flagella-dependent and independent mechanisms such as swarming, surfing or sliding [37,38,39,40,41]. Swarming and surfing are flagella-driven motilities, whereas sliding is a passive appendage-independent bacterial translocation. Swarming is the fastest known type of bacterial motility on surfaces, and it is characterized by the coordinated multicellular migration of cells. This type of motility has been described in several rhizobial species [29,33,42,43,44,45,46,47]. In contrast to swarming, surfing and sliding have only been described in the alfalfa symbiont *Sinorhizobium meliloti* [33,45,48]. Surface spreading in surfing is driven by flagellar rotation and physical/chemical effects created by a secreted exopolysaccharide (EPS), whereas sliding bacterial translocation is powered by the expansive forces of cell growth and facilitated by the production of surfactants that reduce surface tension.

Bacterial translocation on surfaces is greatly influenced by numerous environmental factors, which makes it a complex behaviour to study [40]. In rhizobia, knowledge about the regulatory mechanisms and environmental cues that control surface motility is still limited. Specifically, studies that investigate the influence of root exudates on the ability of rhizobia to spread on surfaces are scarce [43,44,49]. Noteworthy, it has been proposed that the presence of strigolactones (phytohormones that regulate different developmental processes) in plant root exudates promote bacterial surface translocation in some rhizobial species, such as *S. meliloti* GR4 and *Rhizobium leguminosarum* bv. *viciae* [50,51].

In several soil bacteria, biofilm formation and surface motility are inversely regulated [22,25]. Since *nod* gene-inducing flavonoids and NodD1 negatively affect EPS production and biofilm formation in *S. fredii* HH103, in this work we wanted to investigate whether the presence of these molecules also had an effect on *S. fredii* motility. Our data show that the presence of *nod* gene-inducing flavonoids such as genistein stimulates surface translocation in *S. fredii* HH103 but does not affect swimming. The genistein-induced surface motility, which is mediated by both flagella-dependent and -independent mechanisms, is dependent on NodD1, TtsI, and the presence of a functional T3SS. We also show that additional modulators of the *nod* regulon, NolR and MucR, participate in the control of HH103 flavonoid-induced surface motility.

## 2. Results

### 2.1. nod Gene-Inducing Flavonoids Promote Surface Motility in S. fredii HH103

In previous reports, *S. fredii* HH103 showed a slight surface translocation on 0.6% agar semisolid minimal medium (MM) that was promoted by EPS production [52]. Since the presence of *nod* gene-inducing flavonoids reduces EPS production and biofilm formation by HH103 [24], we decided to study whether these compounds also had an effect on HH103 surface motility. In these new experiments, agarose at 0.4% (*w*/*v*) was used as the gelling agent because it increased the reproducibility. Surprisingly, the presence of genistein, an effective inducer flavonoid of HH103 *nod* genes present in soybean root exudates [53], provoked a significant increase in the HH103 surface propagation capacity (76.6 ± 6.6 mm) in comparison to that exhibited in the absence of added compounds (8.2 ± 3.1 mm) on semisolid MM 0.4% agarose after 72 h (Figure 1). In order to investigate whether this induction was specific for genistein or was also promoted by other *nod* gene-inducing flavonoids, we carried out surface motility experiments in the presence of two other effective inducers of HH103, coumestrol and apigenin, and three flavonoids unable to induce *nod* gene expression in this strain: flavanone, 6-hydroxyflavanone, and 5-hydroxyflavone [53]. The results obtained showed a significant increase in HH103 surface translocation in the presence of either coumestrol or apigenin, 64.1 ± 8.4 mm and 68.5 ± 6.4 mm, respectively, although these increases were significantly lower than that obtained in the presence of genistein. In contrast, the surface spreading areas obtained in the presence of non-inducing flavonoids were indistinguishable from those observed under control conditions (ethanol; 8.2 ± 4.7 mm) or in the absence of any added compound to MM (8.2 ± 3.1 mm): 10.5 ± 6.4 mm for flavonone, 8.8 ± 5.2 mm for 6-hydroxyflavanone, and 9.8 ± 5.7 for 5-hydroxyflavone (Figure 1).

### 2.2. The HH103 Genistein-Induced Surface Motility Is Mediated by Flagella-Dependent and -Independent Mechanisms

Bacterial surface motility can be mediated by flagella-dependent or flagella-independent mechanisms [38,41]. In order to elucidate whether the genistein-induced surface spreading exhibited by *S. fredii* HH103 is a flagella-dependent movement, we carried out surface motility experiments with a flagella-less derivative, in which the *flaCBAD* genes were deleted [52]. The results showed that the surface propagation in the presence of the genistein of the Δ*flaCBAD* derivative (45.4 ± 7.8 mm) was reduced by almost 40% when compared to the wild type, indicating that the *S. fredii* HH103 genistein-induced surface motility is the result of a combination of both flagella- dependent and independent surface motilities (Figure 2).

### 2.3. The HH103 Genistein-Induced Surface Motility Depends on the Regulators NodD1, TtsI and NolR

Since the *S. fredii* HH103 surface motility was specifically promoted by *nod* gene-inducing flavonoids, we decided to investigate whether the regulatory proteins involved in the control of *nod* gene expression had a role in this motility. For this purpose, we analysed the surface propagation in response to genistein of HH103 mutants affected in different symbiotic regulators: NodD1, the activator of the *nod* regulon in response to the presence of flavonoids; TtsI, which is the transcriptional activator of the symbiotic T3SS; and different regulatory proteins that take part in the fine-tuning of the *nod* regulon: NodD2, SyrM, NolR and MucR1 [16,17,19,20,21,53]. More specifically, the HH103 mutants examined were: *nodD1*::Ω [53], *nolR*::*lacZ*Δp-Gm^R^ [24], *nodD2*::Ω [20], Δ*mucR1* [19], Δ*syrM* [21], and a HH103 derivative in which the Ω interposon was inserted into the *nod* box that regulates the genistein-induced expression of *ttsI* (indicated as *PttsI*) [17]. In addition, in order to check whether NF could be involved in the genistein-induced surface motility, we included a *nodA* mutant unable to produce these signal molecules [54].

In the absence of flavonoids, and similarly to the wild-type strain, none of the mutants tested exhibited surface motility (Figure 2). However, the results obtained suggested that the genistein-induced surface spreading exhibited by *S. fredii* HH103 is dependent on the NodD1, TtsI and NolR regulatory proteins, since the lack of any of them provoked a severe reduction in genistein-induced surface motility or a complete abolition in the case of the *nolR* mutant (*nodD1* 19.5 ± 9.5 mm; *ttsI* 19.4 ± 5.3 mm; *nolR* 6.6 ± 1.1 mm; Figure 2) when compared to the wild-type strain (78.8 ± 3.8 mm). In contrast, the HH103 *nodD2* and *syrM* mutants did not show significant differences in their propagation levels in comparison to the wild type (79.4 ± 6.8 mm and 74.8 ± 9.6 mm, respectively, Figure 2). An intermediate degree of genistein-induced surface translocation was observed in the *mucR1* mutant (37.6 ± 9.8 mm). The *nodA* mutant exhibited a slight but significant reduction (66.8 ± 12.6 mm; *p* = 0.05) of this motility, suggesting that NF can influence HH103 surface propagation ability.

In order to discard that the impaired surface motility exhibited by some of the mutants could be caused by growth defects on the medium employed for the surface motility experiments, growth curves of the wild-type strain and the different mutants were performed in liquid MM. As shown in Appendix A, none of the mutants assayed exhibited significant differences in growth when compared to the wild-type strain.

The participation of NodD1, NolR, MucR, and TtsI in the genistein-induced surface translocation exhibited by HH103 was confirmed by analysing the behaviour of additional independent mutants in the corresponding regulatory genes: *nodD1*::*lacZ*Δp-Gm^R^ [55], *nolR*::Ω [17], *mucR1*::*lacZ*Δp-Gm^R^ [19], and *ttsI*::Ω [17]. The genistein-induced surface spreading ability of all these mutants (Appendix A) was similar to that of the previously analysed mutants and, therefore, significantly reduced when compared to that of the wild-type strain: 11.2 ± 0.8 for *nodD1*::*lacZ*Δp-Gm^R^, 6.3 ± 1.8 mm for *nolR*::Ω, 40.8 ± 9.0 for *mucR1*::*lacZ*Δp-Gm^R^, and 12.6 ± 4.1 mm for *ttsI*:.Ω.

### 2.4. S. fredii HH103 Swimming Motility Is Not Affected by Genistein

Once we confirmed the positive effect of genistein on the ability of *S. fredii* HH103 to move over surfaces, we wanted to investigate whether this inducer flavonoid could also affect another type of motility, swimming. For this purpose, motility experiments on semisolid BM 0.3% agar in the presence of either genistein or ethanol were performed. As shown in Figure 3, the swimming motility exhibited by HH103 at 72 h upon genistein or ethanol treatment were similar, 14.2 ± 1.2 mm and 14.5 ± 3.6 mm, respectively. In relation to this result, we checked whether any of the HH103 mutants affected in different symbiotic regulators might show changes in swimming motility behaviour in both conditions. The *S. fredii* HH103 Δ*flaCBAD* derivative was also included as a non-motile control. Except for the latter strain (4.5 ± 0.5 mm), no significant differences in the swimming rings were obtained among the different mutants tested in comparison to those of the wild type in the presence of both genistein and ethanol (Figure 3).

Since genistein and different symbiotic regulatory proteins influence the surface motility of *S. fredii* HH103, we also investigated whether they affect flagella production in this bacterium by using transmission electron microscopy. The samples for TEM studies were taken from surface motility assays at 24 h in the presence of either genistein or ethanol. The HH103 Δ*flaCBAD* mutant was used as a negative control of the presence of flagella. The results obtained revealed that the presence or absence of genistein apparently does not affect flagellum production by *S. fredii* HH103 (Figure 4). All the mutants tested showed flagella and no clear differences could be scored when compared to the wild-type strain (Figure 4). These results, together with the swimming motility data, suggest that neither genistein nor the lack of NodD1, NolR or TtsI have a clear effect on flagellum production.

### 2.5. A Functional T3SS Is Required for the S. fredii HH103 Genistein-Induced Surface Motility

As mentioned above, two independent mutants affected in the *ttsI* gene exhibited a dramatic reduction in the genistein-induced surface motility of *S. fredii* HH103, suggesting a role of the HH103 T3SS in this kind of motility. Because of this, we tested the surface motility of HH103 mutants affected in different genes that code for structural components of the *S. fredii* HH103 T3SS: RhcJ (=NolT), an inner MS ring protein; RhcV (=NolV), a structural protein anchored to the inner membrane; and NopA, the T3SS pilus subunit [14,17]. The mutants analysed were *rhcJ*::Tn5-*lacZ* [56], *rhcV*::Ω [57] and *nopA*::*lacZ*Δp-Gm^R^ [17]. All these mutants showed reduced surface motility when compared to HH103 (79.0 ± 3.6 mm). Surprisingly, the *rhcJ* and *rhcV* mutants revealed a more severe phenotype (5.0 ± 1.1 and 7.0 ± 2.4 mm, respectively) than those of the *ttsI* mutants while the *nopA* mutant exhibited similar surface motility to those of *ttsI* mutants (13.8 ± 8.7) (Figure 5). A second, independent, mutant in the *rhcJ* gene (*rhcJ*::Ω; [58]) behaved as the *rhcJ*:: Tn5-*lacZ* mutant (4.9 ± 1.8 mm, Figure 5).

### 2.6. Analysing the Putative Role of HH103 Effector Proteins in Genistein-Induced Surface Motility

The results mentioned above clearly indicated the essentiality of the HH103 T3SS for genistein-induced surface motility. We wanted to investigate whether the involvement of the HH103 T3SS in genistein-induced surface motility is due to a specific effector delivered by this apparatus or to the apparatus itself. For this purpose, we analysed this kind of motility in a collection of mutants affected in each one of the eight different effectors identified in this bacterium so far [58,59,60,61,62,63]: Δ*gunA*, *nopC*::*lacZ*Δp-Gm^R^, *nopD*::Ω, *nopI*::Ω, *nopL*::Ω, Δ*nopM1*-*nopM2*::*lacZ*Δp-Gm^R^ (named as *nopM1-nopM2*), *nopP*:: *lacZ*Δp-Gm^R^, and *nopT*::Ω.

The *S. fredii* HH103 mutants affected in the *gunA*, *nopD*, *nopI*, *nopL*, *nopM1*/*nopM2*, *nopP,* and *nopT* genes showed surface motilities at 72 h upon induction with genistein (79.2 ± 2.2, 81.2 ± 1.1, 79.3 ± 1.0, 79.5 ± 2.5, 75.3 ± 3.1, 76.9 ± 5.5, and 73.2 ± 9.7 mm, respectively) that were undistinguishable from that exhibited by the parental strain HH103 (78.6 ± 3.8 mm) (Figure 6). The *nopC* mutant, instead, showed a significant decrease in the spreading behaviour (42.6 ± 24.7 mm), suggesting an involvement of the NopC effector in *S. fredii* HH103 surface motility.

## 3. Discussion

In this work, we clearly show that surface motility, but not swimming, is regulated in a coordinated manner with the *nod* regulon in *S. fredii* HH103. Surface translocation in this rhizobial strain is triggered by the same flavonoids that activate *nod* gene expression [53] and requires the participation of the positive regulator of the T3SS, TtsI and, because of this, also that of NodD1, which activates the expression of *ttsI* in the presence of *nod* gene-inducing flavonoids [17]. Flavonoids are a large family of phenolic compounds produced by plants where they fulfil a wide variety of functions [64]. One of them is to participate in the plant defence against microbial infections since these compounds are potent antimicrobial agents [64]. In addition to their antibacterial activity, flavonoids may influence different bacterial traits. Thus, these compounds may, depending on the bacterial species, either inhibit or promote biofilm formation by altering the production of extracellular matrix components [65], whereas in *Pseudomonas syringae* pv. tomato DC3000 provoke loss of flagella (reducing swimming and swarming) and inhibit expression and assembly of T3SS [66]. Regarding rhizobia, the long-assumed role of flavonoids as chemoattractants for rhizobia has recently been questioned [67]. To the best of our knowledge, our work is the first study showing an involvement of flavonoids in the induction of surface motility in bacteria.

We found that both the flagellum and the T3SS participate in the flavonoid-induced surface motility exhibited by HH103. Whereas the lack of flagella only provokes a reduction in the level of surface spreading, the inactivation of genes coding for components of the T3SS apparatus results in the complete abolition of the genistein-induced surface motility. Interestingly, the analysis of the HH103 transcriptome upon treatment with genistein has revealed a putative operon containing a flagella-related gene (*flgJ*) and two open reading frames (ORFs) that code for hypothetical proteins whose expression is induced by genistein in a TtsI- and NodD1-dependent manner [18]. We are currently studying whether these three genes are involved in the genistein-induced surface motility exhibited by HH103.

The involvement of the T3SS in HH103 surface motility appears to be related to the apparatus itself rather than to the different effectors secreted by this system. In fact, individual mutants in seven out of the eight different effectors identified in HH103 so far exhibited genistein-induced surface spreading abilities that were undistinguishable from that of the wild-type strain. Regarding NopC, its absence negatively affected this motility although did not provoke the full impairment that was observed in the absence of the T3SS. This protein, which is rhizobial specific, has been traditionally considered as part of the T3SS apparatus [14]. In fact, its coding gene forms part of one operon that also contains the *nopA* and *nopB* genes, both encoding structural proteins of the T3SS pilus [68]. Although it has been recently demonstrated that *S. fredii* HH103 NopC is delivered into host plant cells, suggesting that it carries out a role inside the host [61], the fact that this protein does not contain any known domain highlights that its role remains unknown. Because of all these reasons, it is not possible, at the moment, to elucidate the role of NopC in HH 103 surface motility. Interestingly, the effects of *S. fredii* HH103 *nopC* inactivation on symbiosis (negative with soybean, positive with *Lotus japonicus* Gifu) are similar, although at a lower extent, to those of the absence of the T3SS [61,63], which is the same tendency that we have observed for genistein-induced surface motility. In summary, the existing data indicate that NopC plays an important, but not essential, role in the proper functionality of the HH103 T3SS. It is also important to mention that the less drastic phenotype exhibited by mutants affected in *ttsI* when compared to some structural T3SS mutants (*rhcJ*, *rhcV*) might be due to the basal expression of T3SS related genes that has been observed in the absence of TtsI [18].

In different animal pathogenic bacteria, T3SS gene expression is positively correlated with swarming motility, and several environmental cues have been identified that regulate synthesis of the secretion apparatus and surface translocation in a coordinated manner [69,70,71]. Our investigations show for the first time in a rhizobial strain that specific plant flavonoids act as a cue that can activate both T3SS synthesis and surface motility. Interestingly, the T3SS of the animal pathogen *Shigella flexneri* has been involved in the secretion of a biosurfactant-like molecule that helps the bacteria to invade epithelial cells but also facilitates swarming motility [72]. The possibility that the T3SS of HH103 participates in the secretion of a compound with surfactant properties that contributes to surface spreading in this rhizobial strain warrants further investigations.

In addition to NodD1 and TtsI, two global regulators related to the *nod* regulon also participate in the control of HH103 surface motility: *nolR* and *mucR1* [16,19,20]. NolR regulates the expression of hundreds of genes and, regarding symbiosis, has an opposite role to that of NodD1 [16,20,24]: it represses the expression of genes related to NF production and Nops secretion (thus reducing the production of these two kind of molecular signals) but enhances the production of EPS, that in *S. fredii* is negatively regulated by *nod* gene-inducing flavonoids in a NodD1- and SyrM-dependent manner. Thus, interestingly, surface motility is the first HH103 trait in which a similar effect (positive in this case) of NodD1 and NolR has been found. Transcriptomic studies of the lack of a functional *nolR* gene have been carried out upon treatment with *Lotus japonicus* Gifu root exudates [20]. In addition to increased expression of genes involved in NF production as well as of many genes involved in Nops secretion in the *nolR* background, these studies revealed differential expression of several genes involved in flagella synthesis. Thus, *flgG*, *fliN*, *fliG flgD*, *flgE*, and *flgL* were underexpressed (−10.4, −6.0, −4.1, −3.4, −3.3, −2.0, respectively) whereas *flaD* and *flgJ* (2.0 and 2.9) were slightly overexpressed in a *nolR* mutant with regard to the parental strain HH103 [20]. These changes in flagellar gene expression might have an impact in flagellum functioning in surface spreading. On the other hand, we have previously demonstrated that HH103 NolR is required for EPS production, which might also influence surface motility [16]. Thus, more research is required to unveil the exact mechanism by which NolR affects HH103 genistein-induced surface motility but, in any case, a lack of NolR does not prevent flagellum synthesis in HH103.

It is known that MucR1 or its *Rhizobium leguminosarum* orthologue, RosR, plays a global regulator role and takes part in numerous processes in free-living and bacteroids cells [19,73,74,75,76]. Its mutation provokes a pleiotropic phenotype, and in the case of *R. leguminosarum* causes swimming and surface motility alterations [74,75]. A transcriptomic analysis of a *S. fredii* HH103 *mucR1* mutant revealed that a high number of flagella-related genes showed overexpression both in the absence and in the presence of genistein when compared to the parental strain [19]. In a previous work in which more restrictive conditions for surface spreading were used, no differences in this motility were scored between an *mucR* mutant and the wild-type strain [19]. However, in this work, in which conditions to analyse HH103 surface motility have been optimised, we show that the inactivation of *mucR1* caused a partial impairment in genistein-induced surface motility. Since the surface motility exhibited by HH103 is due to both flagella-dependent and -independent mechanisms, there are different possibilities that should be checked to understand the *mucR1* mutant behaviour. On the one hand, the overexpression of several *flg*, *fli*, *fla*, and *mot* genes caused by the lack of a functional *mucR1* gene might be detrimental for the flagella-dependent mechanism. On the other hand, the absence of MucR1 might be detrimental for the flagella-independent mechanism. In addition, similarly to NolR, HH103 MucR1 is a positive regulator of EPS production [19]. Clearly, more research is needed to shed light on this issue but, as in the case of NolR, the HH103 flagellum is assembled in the absence of a functional MucR1 protein.

Another interesting result obtained in this work is the fact that a HH103 *nodA* mutant, unable to produce NF, exhibited a partial impairment in genistein-induced surface motility. Interestingly, a previous work of our group showed that the inactivation of the *nodA* gene of HH103 altered EPS production and reduced biofilm formation in the presence of genistein [24]. In addition, *S. meliloti* Nod factors have been shown to be important for biofilm formation by this bacterium [77]. Whether the negative effect of the mutation of *nodA* on HH103 surface motility is related to the absence of NF, to the alteration of EPS production and biofilm formation, or to other unidentified consequence remains to be elucidated.

Bacterial motility is not crucial for the establishment of rhizobia–legume symbiosis, but it affects the ability of rhizobial strains to quickly colonise and infect host legume roots (revised by [28]), which might be an important feature in natural conditions where different rhizobial strains can compete for nodulating the same plant. Unfortunately, all the HH103 mutants tested in this study and shown to be impaired in surface motility are affected in genes with relevant symbiotic functions [16,17,19,20,53,61], which prevents gaining any knowledge about the putative relevance of genistein-induced surface motility for HH103 symbiotic capabilities. As mentioned before, currently we are studying several HH103 genes that might be related to this kind of motility and that might shed some light about the possibility that genistein-induced surface motility may have symbiotic relevance.

In summary, we show in this work that surface motility in *S. fredii* HH103 is under the control of the early signalling events that take place in the rhizobia–legume symbiotic interaction. *nod* gene-inducing flavonoids and NodD1 activate the expression of the T3SS through TtsI. This apparatus not only delivers effector proteins in host cells but also is absolutely essential for surface spreading of this rhizobial strain. To the best of our knowledge, this is the first time that flavonoids and symbiotic transcriptional regulators have been demonstrated to induce surface motility in a rhizobial strain. In addition, the bacterial flagellum, and maybe NF, is necessary for full surface motility ability. The genetic control of this motility is complex and involves the participation of additional symbiotic transcriptional regulators such as NolR and MucR1. Future works may elucidate the exact roles that different structures (T3SS and the flagellum) and molecules (NF) play in this kind of motility.

## 4. Materials and Methods

### 4.1. Basic Molecular and Microbiological Techniques

*Sinorhizobium fredii* HH103 Rif^R^ [78] and its mutant derivatives (listed in Appendix A) were grown at 28 °C on TY medium [79], Bromfield medium (BM) (0.04% tryptone, 0.01% yeast extract, and 0.01% CaCl_2_ 2H_2_O) [80] or minimal medium (MM) containing glutamate (6.5 mM), mannitol (55 mM), mineral salts (1.3 mM K_2_HPO_4_, 2.2 mM KH_2_PO_4_·3H_2_O, 0.6 mM MgSO_4_·7H_2_O, 0.34 mM CaCl_2_·2H_2_O, 0.022 mM FeCl_3_·6H_2_O, 0.86 mM NaCl), and vitamins (0.2 mg/L biotin, 0.1 mg/L calcium pantothenate) [81]. When required, the media were supplemented with the appropriate antibiotics as described previously [53]. Flavonoids were prepared in ethanol as 1000-fold concentrated solutions and used at a final concentration of 1 μg/mL (which corresponds to 3.7 µM for the case of genistein). Control treatments contained 0.1% ethanol.

To improve reproducibility, all liquid cultures of *S. fredii* HH103 and its derivatives were routinely initiated from glycerol stocks. The ability of HH103 and different mutant derivatives to grow in liquid MM was monitored every 2 h in a Genesys 20 visible spectrophotometer (ThermoFisher Scientific, Waltham, MA, USA). For this purpose, starter cultures of each strain were grown in the corresponding medium until late exponential phase (OD_600_ = 1.0), and 4 and 40 µL of the starters were used to inoculate 5-mL of medium.

### 4.2. Motility Assays

Surface motility was analysed on semisolid MM plates containing 0.4% agarose and, when corresponds, supplemented with ethanol or a flavonoid (at final concentrations of 0.1% *v*/*v* and 1 μg/mL, respectively). Plates were prepared according to a rigorous protocol [47] since the water content of the medium is a crucial factor for surface motility. Each plate contained 20 mL of medium that was poured when the agar was relatively cool (∼50 °C). After pouring, open plates were dried in a laminar flow hood for 15 min. For inoculation, cells from 1 mL of culture grown in TY broth to the late exponential phase (OD_600nm_ = 1) were washed two times with MM and 10-fold concentrated. Every plate was inoculated on top with 2 µL of this bacterial suspension. After inoculation, plates remained open, and the culture drops were allowed to dry in a laminar flow hood for 10 min. Finally, plates were incubated at 28 °C for 72 h, and surface motility was scored every 24 h. Only results scored at 72 h are presented.

Swimming was examined on plates prepared with BM containing 0.3% Bacto agar with ethanol or genistein. 3 µL aliquots of rhizobial cultures grown in TY (OD_600nm_ = 1) were inoculated into these plates.

The migration zone was determined as the colony diameter (mm) after 72 h (for swimming and surface motility) of incubation. In the case of surface motility experiments performed on semisolid MM, in which fractal patterns with characteristic tendrils were formed, migration zones were calculated as the average length of the two sides of a rectangle able to exactly frame each colony. Swimming and surface motility experiments were performed at least with two and three biological replicates, respectively, and three technical replicates each one.

### 4.3. Statistical Analyses of Motility Data

In the experiment shown in Figure 1, each specific condition was individually compared to the control condition (MM). In the experiments shown in Figure 2, Figure 3, Figure 5 and Figure 6, data obtained for each strain were compared to those of the wild-type strain HH103 cultured in the same condition. In all cases, statistical comparisons were performed by using the non-parametric test of Mann–Whitney, using a *p* value of 0.01 for considering differences as statistically significant.

### 4.4. Transmission Electron Microscopy (TEM)

Cells for TEM observations were obtained from the edge of colonies grown on the surface of MM after 24 h of incubation in the presence of genistein or ethanol at 28 °C. Carbon-coated Formvar grids were placed for 5 min on top of a drop of water previously applied to the colony border. The grids were then washed twice in water for 1 min and stained with 2% (*w*/*v*) uranyl acetate for 3 min. Grids were allowed to air dry for at least 1 h and visualised using a Zeiss Libra 120 TEM a 100 kV beam at the Microscopy Service in Centro de Investigación, Tecnología e Innovación (CITIUS, University of Seville, Seville, Spain).

## Figures and Tables

**Figure 1 ijms-23-07698-f001:**
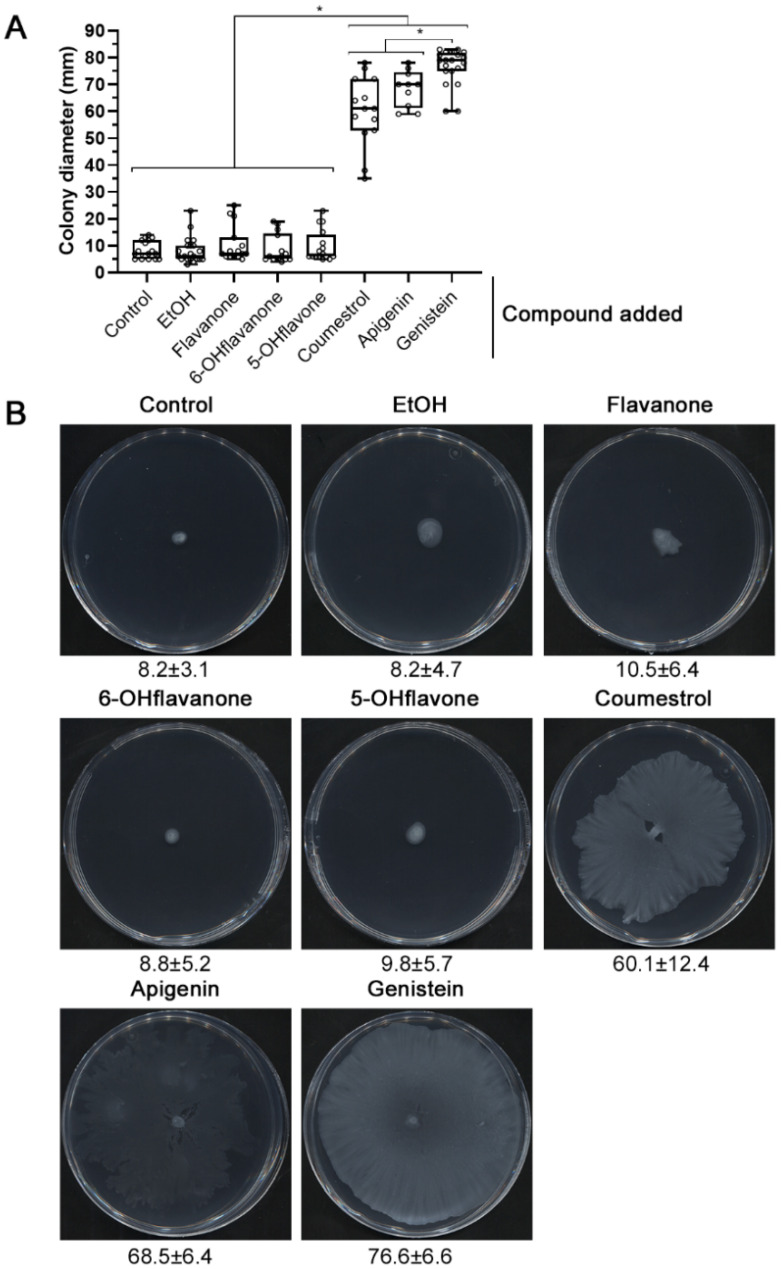
Effect of different compounds (including *nod* gene-inducing and non-inducing flavonoids) on *S. fredii* HH103 surface motility. Surface motility experiments were performed on minimal medium (MM) using 0.4% agarose as the gelling agent and scored at 72 h. (**A**). Box and whisker plots from at least three biological replicates performed with three technical replicates. Asterisks (*) indicate significant differences with the corresponding control sample using the non-parametric test of Mann–Whitney, α = 1%. (**B**). Representative pictures of the motilities exhibited by *S. fredii* HH103 under each condition. Values under images represent the average and standard deviation of surface migration (given in millimetres and determined as described in the text).

**Figure 2 ijms-23-07698-f002:**
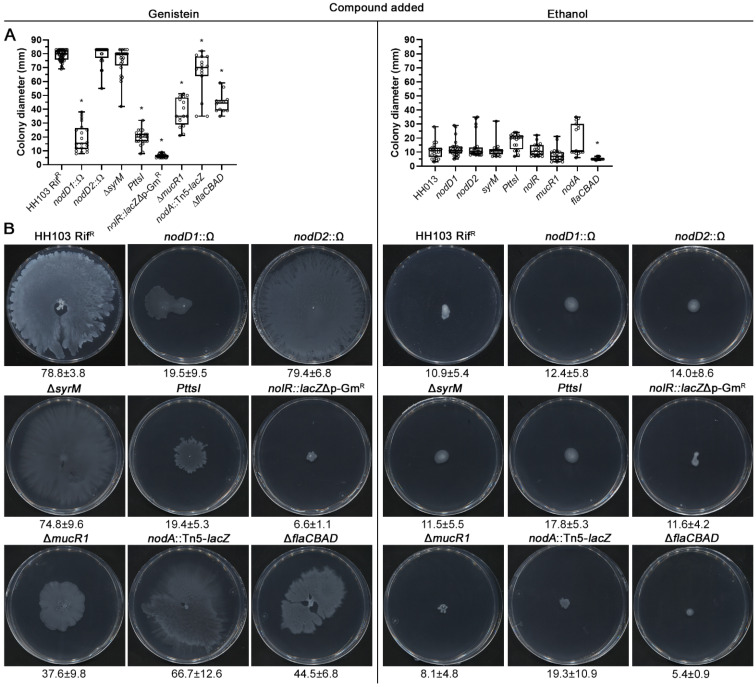
Role of different symbiotic regulators on the *S. fredii* HH103 surface translocation in the presence of genistein. The non-flagellated (*flaCBAD*) and *nodA* mutants were also included. Surface motility experiments were performed on minimal medium (MM) using 0.4% agarose as gelling agent and scored at 72 h. (**A**). Box and whisker plots from at least three biological replicates performed with three technical replicates. Asterisks (*) indicate significant differences with the corresponding control sample using the non-parametric test of Mann–Whitney, α = 1%. (**B**). Representative pictures of the motilities exhibited in the presence of genistein by the different *S. fredii* HH103 mutants analysed. Values under images represent the average and standard deviation of surface migration (given in millimetres and determined as described in the text).

**Figure 3 ijms-23-07698-f003:**
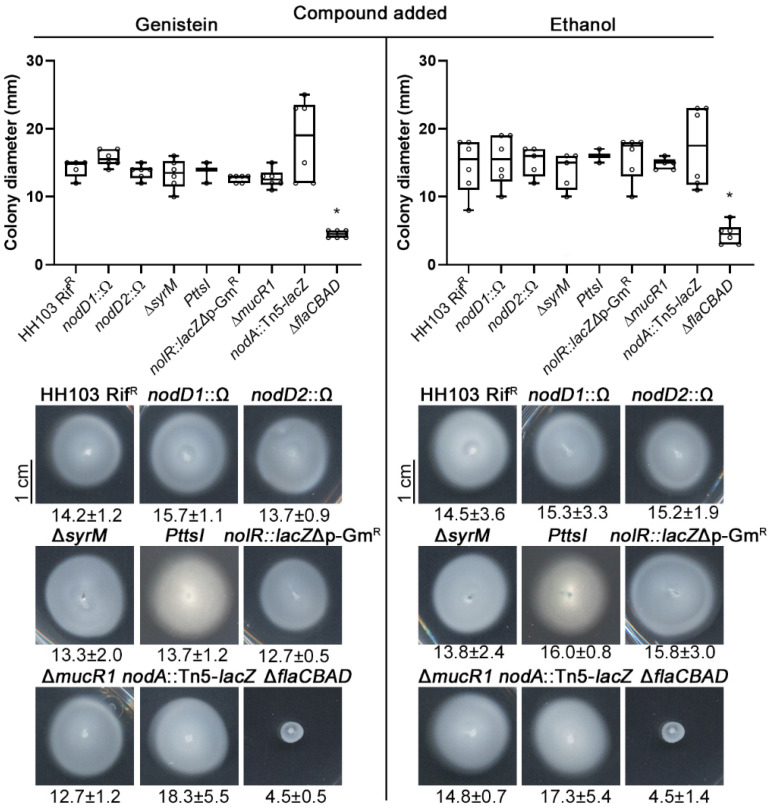
Swimming abilities of *S.*
*fredii* HH103 and different mutants in the absence and the presence of genistein at 72 h. All data points are shown in box and whisker plots from at least three biological replicates performed with three technical replicates. Asterisks (*) indicate significant differences with the corresponding control sample using the non-parametric test of Mann–Whitney, α = 1%. Representative pictures of the motilities exhibited in the presence of genistein by the different *S. fredii* HH103 mutants analysed are also shown. Values under images represent the average and standard deviation of migration (given in millimetres and determined as described in the text).

**Figure 4 ijms-23-07698-f004:**
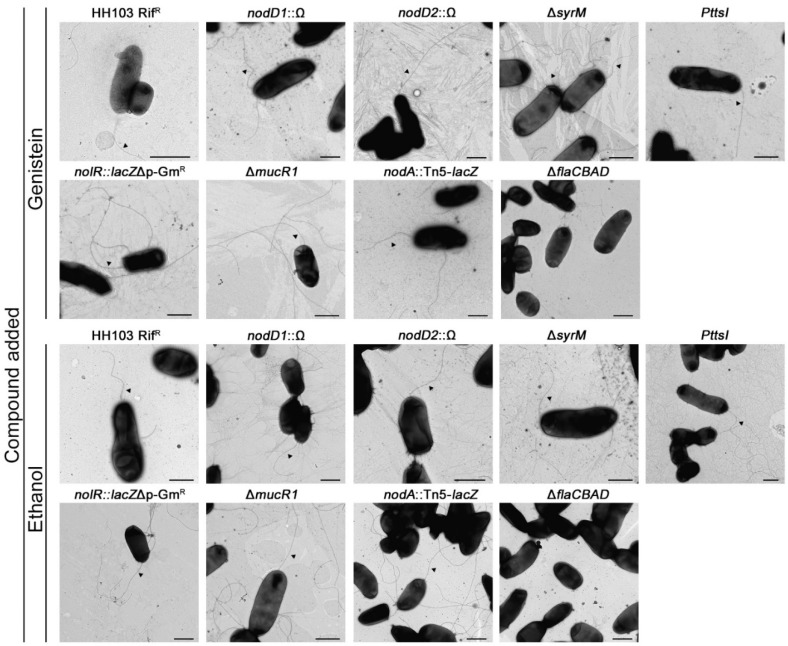
Transmission electron microscope (TEM) images of *S. fredii* HH103 and different mutants. Cells were isolated from the edge of colonies grown on semisolid MM 0.4% agarose in the presence of genistein or ethanol after 24 h of incubation at 28 °C and stained with 2% uranyl acetate. Scale bars correspond to 1 µm.

**Figure 5 ijms-23-07698-f005:**
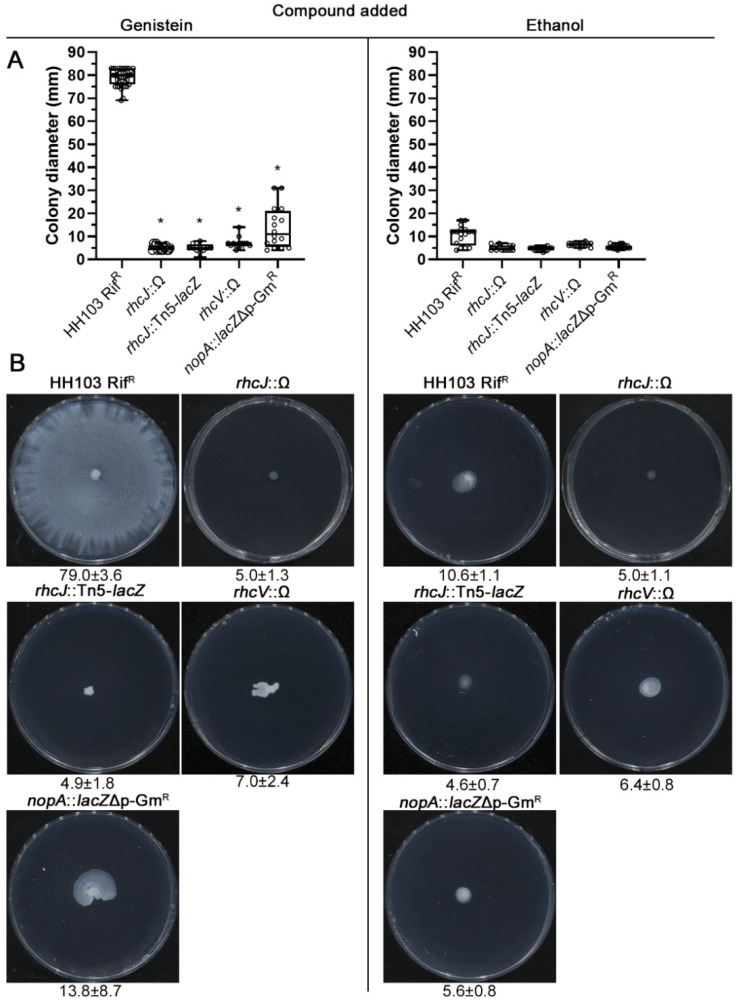
Role of different components of the T3SS apparatus on the *S. fredii* HH103 surface translocation in the presence of genistein. Surface motility experiments were performed on minimal medium (MM) using 0.4% agarose as gelling agent and scored at 72 h. (**A**). Box and whisker plots from at least three biological replicates performed with three technical replicates. Asterisks (*) indicate significant differences with the corresponding control sample using the non-parametric test of Mann–Whitney, α = 1%. (**B**). Representative pictures of the motilities exhibited in the presence of genistein by the different *S. fredii* HH103 mutants analysed. Values under images represent the average and standard deviation of surface migration (given in millimetres and determined as described in the text).

**Figure 6 ijms-23-07698-f006:**
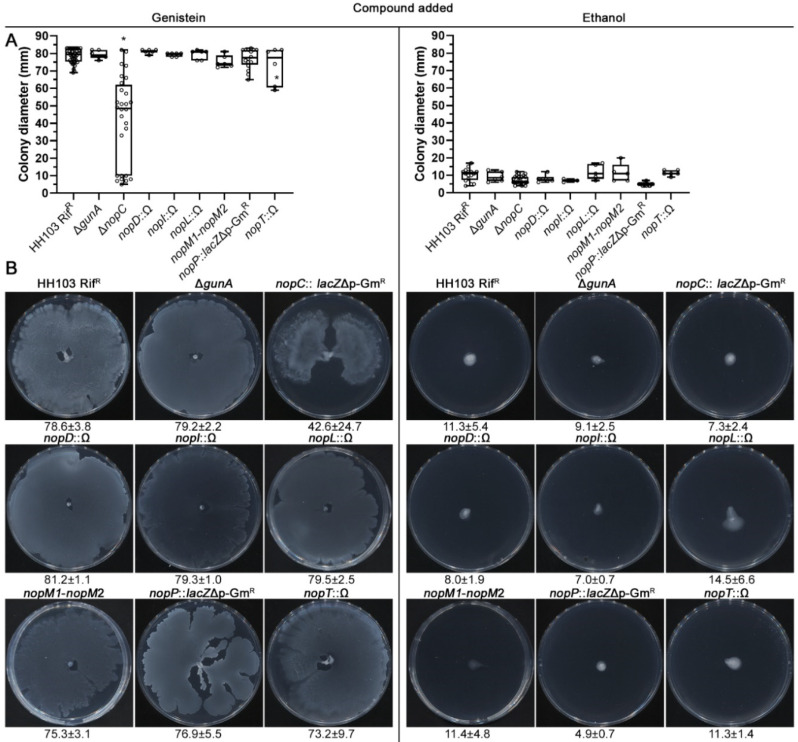
Role of the different T3SS effectors on the *S. fredii* HH103 surface translocation in the presence of genistein. Surface motility experiments were performed on minimal medium (MM) using 0.4% agarose as gelling agent and scored at 72 h. (**A**). Box and whisker plots from at least three biological replicates performed with three technical replicates. Asterisks (*) indicate significant differences with the corresponding control sample using the non-parametric test of Mann–Whitney, α = 1%. (**B**). Representative pictures of the motilities exhibited in the presence of genistein by the different *S. fredii* HH103 mutants analysed. Values under images represent the average and standard deviation of surface migration (given in millimetres and determined as described in the text).

## Data Availability

Not applicable.

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
