# Peer review of "Surface Motility Regulation of Sinorhizobium fredii HH103 by Plant Flavonoids and the NodD1, TtsI, NolR, and MucR1 Symbiotic Bacterial Regulators"

_ijms, 2022, doi:10.3390/ijms23147698_

Round 1

Reviewer 1 Report

  • The aim of the study has to be staged in “Introduction”.
  • The last part of the “Introduction” better fits the summary of the “Discussion”.
  • The “Materials and Methods” have to be supplemented with a description of statistical analysis with particular emphasis on the determination of statistical significance.
  • Please consider the collaboration with some good statistician, which could improve the statistical evaluation and allow to draw more scientific valuable conclusions from the conducted research.
  • In my opinion, data presented in Supplementary Materials Figure 3S should be presented in the main part of the manuscript.

 Some sentences have to be rewritten because they are impropriety in the case of scientific papers eg.

Lines 527-529 “Because of this, it is not possible, at the moment, to elucidate the role of NopC in surface motility, although it is clear that it is not essential”

Lines 532-534 “All these results point out to an important, but not essential, role of NopC in the proper functionality of the HH103 T3SS.”

Lines 543-544 “For obvious reasons, there are not many data in the literature about the role of T3SS in bacterial surface motility”

Author Response

Reviewer 1
The aim of the study has to be staged in “Introduction”.
Thank you very much for your comment. We have modified the last paragraph of
the Introduction and include a sentence stating the aim of this work (lanes 122-
124 of the tracking-changes version of the manuscript).

The last part of the “Introduction” better fits the summary of the “Discussion”.
In this paragraph we wanted to summarize the main conclusions obtained in this
work. We have modified and reduced this paragraph, and as mentioned above, we
have also included a sentence stating the aim of our work.

The “Materials and Methods” have to be supplemented with a description of
statistical analysis with particular emphasis on the determination of statistical
significance.

Thank you for your suggestion. We have included a new section into Material and
Methods and detailed the statistical analyses that have been carried out in this
work (sub-section 4.3, lanes 739-745 of the tracking-changes version of the
manuscript).

Please consider the collaboration with some good statistician, which could
improve the statistical evaluation and allow to draw more scientific valuable
conclusions from the conducted research.

Thank you very much for your piece of advice. In fact, long time ago we required
de advice of statisticians with expertise in the analysis of biological data about the
statistical test that we should employ for our work. They recommended us the use
of non-parametric tests that do not assume a Gaussian distribution of the data. In
this work we have made individual comparisons of each treatment with the control
by using the non-parametric test of Mann-Whitney.

In my opinion, data presented in Supplementary Materials Figure 3S should be
presented in the main part of the manuscript.

Thank you for your comment. Following your suggestion, original Figure S3 is
now Figure 4 in the revised version of the manuscript.

Some sentences have to be rewritten because they are impropriety in the case of
scientific papers eg.

Lines 527-529 “Because of this, it is not possible, at the moment, to elucidate the role of NopC in surface motility, although it is clear that it is not essential”

This sentence has been corrected (please, see lines
598-600 of the tracking-changes
version of the manuscript).

Lines 532-534 “All these results point out to an important, but not essential, role of NopC
in the proper functionality of the HH103 T3SS.”

This sentence has been corrected (please, see lines 604-605 of the tracking-changes
version of the manuscript).

Lines 543-544 “For obvious reasons, there are not many data in the literature about the
role of T3SS in bacterial surface motility”

This sentence has been removed from the manuscript
.

Reviewer 2 Report

The manuscript “Sinorhizobium fredii HH103 surface motility is regulated by plant flavonoids and the NodD1, TtsI, NolR, and MucR1 symbiotic bacterial regulators” is an interesting study of mode of motility and influencing genes. The study explores the presence of nod gene-inducing flavonoids does not affect swimming but promotes a mode of surface translocation, which involves both flagella-dependent and -independent mechanisms. The surface motility Sinorhizobium fredii HH103 is regulated in a flavonoid-NodD1-TtsI dependent manner, relies on the assembly of the symbiotic type secretion system (T3SS), and involves the participation of additional modulators of the nod regulon (NolR and MucR1).

1.     Title should be modified somewhat like: Surface motility regulation of Sinorhizobium fredii HH103 by plant flavonoids and the NodD1, TtsI, NolR, and MucR1 symbiotic bacterial regulators. Title should not be in sentence format.

2.     “As shown in Figure 3, the swimming motility exhibited by HH103 at 72 h upon genistein or ethanol treatment were similar, 14.2±1.2 mm and 14.5±3.6 mm, respectively." This is quite interesting; however, I wonder that why authors selected ethanol?

3.     First paragraph of discussion seems like repetition of introduction. This paragraph should be rewritten based on the results and related literatures.

Author Response

The manuscript “Sinorhizobium fredii HH103 surface motility is regulated by plant
flavonoids and the NodD1, TtsI, NolR, and MucR1 symbiotic bacterial regulators” is an
interesting study of mode of motility and influencing genes. The study explores the
presence of nod gene-inducing flavonoids does not affect swimming but promotes a mode of surface translocation, which involves both flagella-dependent and -independent mechanisms. The surface motility Sinorhizobium fredii HH103 is regulated in a flavonoid-NodD1-TtsI dependent manner, relies on the assembly of the symbiotic type secretion system (T3SS), and involves the participation of additional modulators of the nod regulon (NolR and MucR1).

1. Title should be modified somewhat like: Surface motility regulation of
Sinorhizobium fredii HH103 by plant flavonoids and the NodD1, TtsI, NolR, and
MucR1 symbiotic bacterial regulators. Title should not be in sentence format.

Thank you for your suggestion. Done.

2. “As shown in Figure 3, the swimming motility exhibited by HH103 at 72 h upon
genistein or ethanol treatment were similar, 14.2±1.2 mm and 14.5±3.6 mm,
respectively." This is quite interesting; however, I wonder that why authors
selected ethanol?

Thank you for your comment. In this work we have used medium supplemented with
ethanol as control because this compound is the solvent used for dissolving
flavonoids. The control media have as much as ethanol as those supplemented
with flavonoids.

3. First paragraph of discussion seems like repetition of introduction. This paragraph
should be rewritten based on the results and related literatures.

Thank you for your comment. We have modified this paragraph accordingly to your
suggestion, trying to avoid redundancy with the Introduction. Please, note that the first 4 lines has been moved to another part of the Discussion (lines 672-675 of the tracking-changes version of the manuscript)